# The Geometry and Topology of Modular Addition Representations

**Gabriela Moisescu-Pareja**[*]
McGill University, Mila

**Gavin McCracken**[*]
McGill University, Mila

**Harley Wiltzer**
McGill University, Mila

**Vincent Létourneau**
Université de Montréal, Mila

**Colin Daniels**
Independent

**Jonathan Love**
Leiden University

## Abstract

The *Clock* and *Pizza* interpretations, associated with neural architectures differing in either uniform or learnable attention, were introduced to argue that different architectural designs can yield distinct circuits for modular addition. Applying geometric and topological analyses to learned representations, we show that this is not the case: Clock and Pizza circuits are topologically and geometrically equivalent and are thus equivalent representations.

## 1 Introduction

Group multiplication tasks such as modular addition have become a standard testbed for toy models in interpretability Nanda et al. [2023], Chughtai et al. [2023], Gromov [2023], Morwani et al. [2024], McCracken et al. [2025a], He et al. [2024], Tao et al. [2025], Doshi et al. [2023], McCracken et al. [2025b]. The task is non-linearly separable yet mathematically well understood, making it ideal for researching how networks internally compute solutions. Two influential works examined modular addition in transformers, finding different architectures give rise to different circuits. Nanda et al. [2023] described a "Clock" interpretation, while Zhong et al. [2023] introduced the contrasting "Pizza" interpretation, each tied to architectural choices. We revisit these claims using geometric and topological analyses and find that Clock and Pizza learn topologically equivalent representations and thus the same circuit. In contrast, our new architecture, MLP-Concat, produces a genuinely different representation and thus a different circuit.

## 2 Background and Setup

We consider various neural network architectures for the task of modular addition, which means predicting the map $(a, b) \mapsto a + b \mod n$ for $a, b \in \mathbb{Z}_n$. For the sake of this paper, we fix $n = 59$. All architectures begin by embedding the inputs $a, b$ to vectors $\mathbf{E}_a, \mathbf{E}_b \in \mathbb{R}^{128}$ using a shared (learnable) embedding matrix. The architectures differ in how the embeddings are then processed: **MLP-Add** immediately passes $\mathbf{E}_a + \mathbf{E}_b$ through an MLP, **MLP-Concat** immediately passes the concatenation $\mathbf{E}_a \oplus \mathbf{E}_b \in \mathbb{R}^{256}$ through an MLP, and **Clock** and **Pizza**, introduced by Zhong et al. [2023] pass $\mathbf{E}_a, \mathbf{E}_b$ through a self-attention layer before the MLP. Particularly, **Pizza** [Zhong et al., 2023] uses a fixed, constant attention matrix, while **Clock** [Nanda et al., 2023] uses the standard scaled softmax attention. We refer to transformer-based architectures associated with the Clock as **Attention 1.0** and those associated with Pizza as **Attention 0.0**, respectively.

---

[*]Equal contribution. `{gabriela.moisescu-pareja, gavin.mccracken}@mail.mcgill.ca`

It is well-known [Nanda et al., 2023, Zhong et al., 2023] that the above architectures learn circuits with learned embeddings of the following form,

$$\mathbf{E}_a = [\cos(2\pi fa/n), \sin(2\pi fa/n)], \quad \mathbf{E}_b = [\cos(2\pi fb/n), \sin(2\pi fb/n)]. \quad (1)$$

What distinguishes them is how the embeddings are *transformed* post-attention. Treating the attention as a blackbox and looking at its output $\mathbf{E}_{ab}$, the two claims follow. **Clock** computes the *angle sum*,

$$\mathbf{E}_{ab} = [\cos(2\pi f(a+b)/n), \sin(2\pi f(a+b)/n)] \quad (2)$$

encoding the modular sum on the uxit circle, which needs second-order interactions (e.g., multiplying embedding components via sigmoidal attention). In **Pizza**, $\mathbf{E}_{ab}$ adds the embeddings directly as $\mathbf{E}_a + \mathbf{E}_b$, giving:

$$\mathbf{E}_{ab} = [\cos(2\pi fa/p) + \cos(2\pi fb/p), \sin(2\pi fa/n) + \sin(2\pi fb/n)], \quad (3)$$

producing a *vector addition* on the circle, which is entirely linear in the embeddings. McCracken et al. [2025a] showed that, across **Clock**, **Pizza**, and **MLP-Concat** architectures, first layer neurons then take the form of so-called *simple-neurons*, producing pre-activations $N(a, b)$ given by

$$N(a, b) = \cos(2\pi fa/p + \phi_a) + \cos(2\pi fb/p + \phi_b), \quad (4)$$

where frequencies $f$ and phases $\phi_a, \phi_b$ are learned across training.

## 3 Methodology

In the simple neuron model the only degrees of freedom beyond the frequency of a neuron are the learned *phases*. We analyze the structure of learned representations using two empirical methods: phase distributions and their induced topological structure. See Appendix A for additional details and computational methodology.

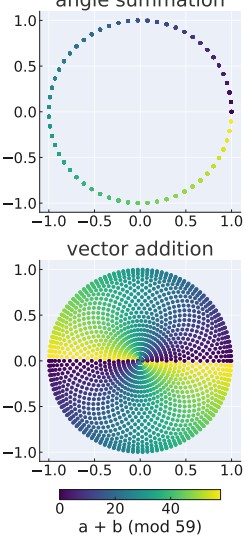

Figure 1: Clock and Pizza's analytical forms. Points are $\mathbf{E}_{ab}$ (cf. (2), (3)), colored by $(a + b)$ mod 59.

**Phase Alignment Distributions.** We propose the *Phase Alignment Distribution* (PAD). To a given architecture, a PAD is a distribution over $\mathbb{Z}_n \times \mathbb{Z}_n$. Samples of this distribution are drawn as follows:

1. Sample a random initialization and train the network, then sample a neuron uniformly.

2. Return pair $(a, b) \in \mathbb{Z}_n \times \mathbb{Z}_n$ achieving the largest activation in the resulting neuron.

A PAD illustrates, across independent training runs and neuron clusters, how often activations are maximized on the $a = b$ diagonal—that is, it depicts how often learned phases align *i.e.* $\phi_a = \phi_b$. Even beyond inspecting the proximity of samples to this diagonal, we propose to compare the PADs of architectures according to metrics on the space of distributions over $\mathbb{Z}_n \times \mathbb{Z}_n$, giving an even more precise comparison. In the following section, we will provide estimates of the PADs for the aforementioned architectures, as well as distances between PADs under the *maximum mean discrepancy* [Gretton et al., 2012, MMD]—a family of metrics with tractable unbiased sample estimators.

**Betti numbers.** *Betti numbers* distinguish the structure of different stages of circuits across layers. The $k$-th Betti number $\beta_k$ of a topological manifold counts $k$-dimensional holes: $\beta_0$ counts connected components, $\beta_1$ counts loops, $\beta_2$ counts voids enclosed by surfaces. For reference, a disc has Betti numbers $(\beta_0, \beta_1, \beta_2) = (1, 0, 0)$, a circle has $(1, 1, 0)$, and a 2-torus has $(1, 2, 1)$. We estimate the distribution over Betti number vectors corresponding to the set of neurons in a given layer to distinguish the structure of the layers.

## 4 Discussion and Conclusion

This work set out to clarify whether the **Clock** and **Pizza** interpretations (corresponding to Attention 1.0 and 0.0 architectures respectively) for modular addition implement distinct circuits or merely

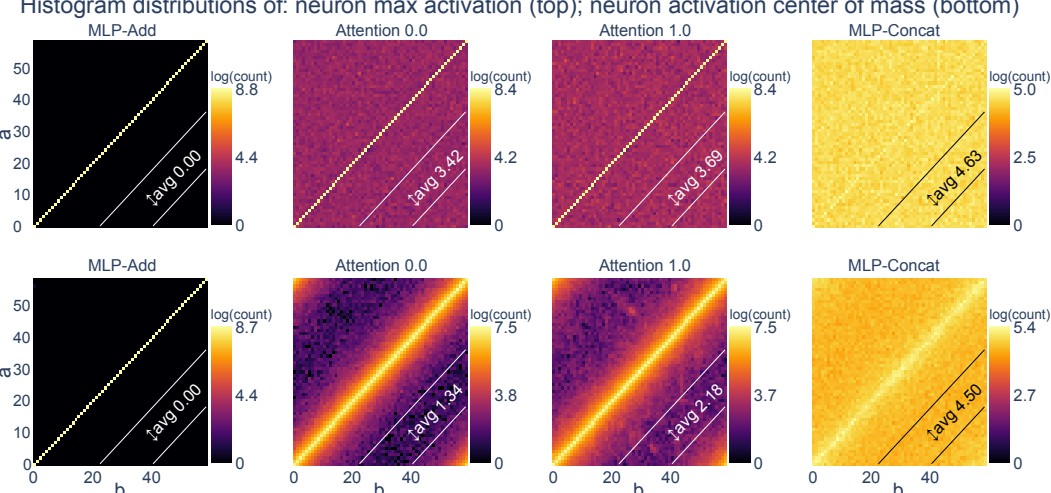

Figure 2: Log-density heatmaps for the distribution of neuron maximum activations (top) and activation center of mass (bottom) across 703 models. Attention 0.0 and 1.0 architectures show modest off-diagonal spread relative to MLP-Add, but remain constrained by architectural bias toward diagonal alignment. MMD scores between Attention 0.0 and 1.0 are 0.0237 (row 1) and 0.0181 (row 2), indicating near identical distributions (see Table 1).

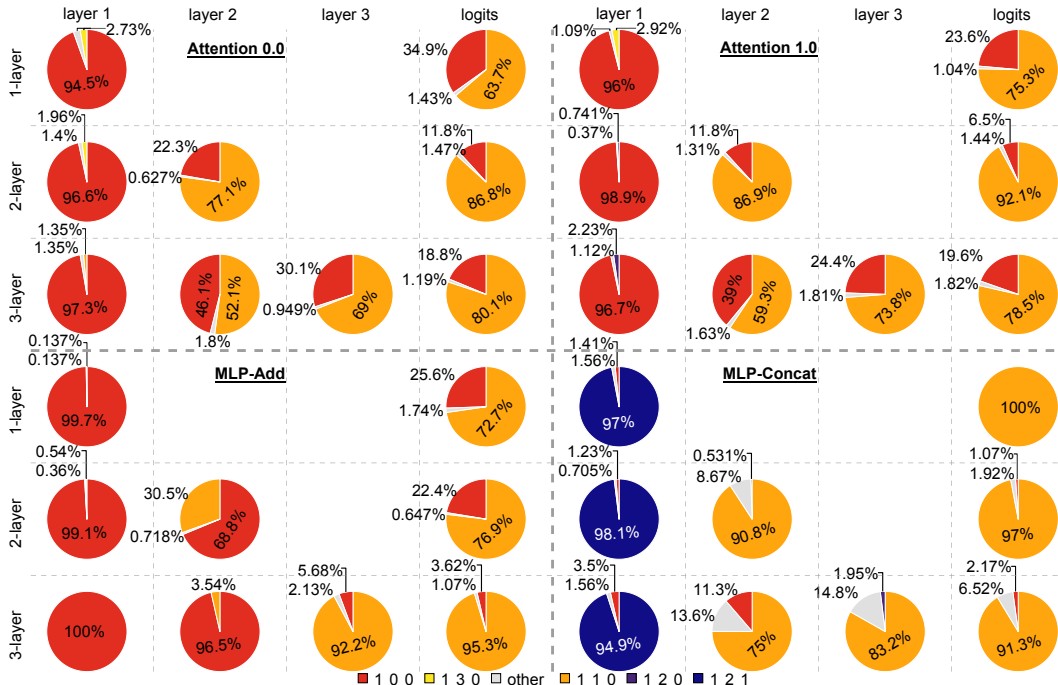

Figure 3: Betti number distributions across layers for 1-, 2-, and 3-layer models (100 seeds for each model). In layer 1, MLP-Add, Attention 0.0, and Attention 1.0 mostly yield disc-like representations, while MLP-Concat produces a torus. From the second layer onward, MLP-Add and both Attention variants converge to either a disc or a circle: the circle reflects the logits topology (correct answer), while the disc is a transient intermediate that can persist in later layers. MLP-Concat instead transitions directly to the circle. Across depth, Attention 0.0 and 1.0 are nearly identical with the latter having fewer transient discs.

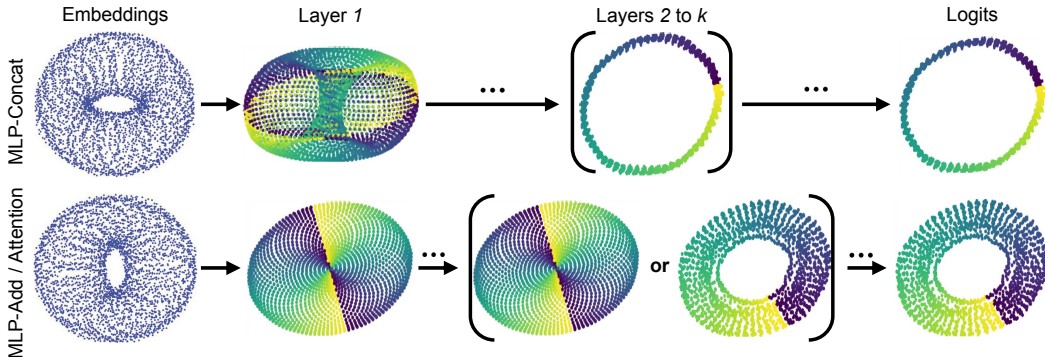

Figure 4: PCA projections of embeddings, intermediate pre-activations, and logits for MLP-Concat (top) and MLP-Add/Attention variants (bottom). In layer 1, MLP-Add and Attention models form discs like vector addition (Fig. 1), while MLP-Concat forms a torus (Fig. 3); in later layers and at logits representations in all models approach circles (angle summation, Fig. 1), but MLP-Concat immediately reaches a thin angle summation circle after layer 1.

reflect superficial differences. Using geometric and topological analyses, we find that their internal representations are in fact highly similar. The PAD analysis (Fig. 2) shows that both architectures produce distributions closely aligned with the $a = b$ diagonal, nearly indistinguishable under MMD (Appendix B for additional experiments and statistical significance). Betti number analysis (Fig. 3) confirms that their topological trajectories across layers converge in the same way, while MLP-Concat follows a different path. Thus, the distinction between "Clock" and "Pizza" is largely illusory: both instantiate the same underlying circuit, differing more from MLP-Concat than from each other and MLP-Add.

More broadly, we show that architectures with trainable embeddings approximate the torus-to-circle map, with differences arising in how this map *factors* through intermediate representations. This perspective connects to the *manifold hypothesis* [Bengio et al., 2013], which posits that networks discover low-dimensional manifolds underlying data. Our results demonstrate how high-level architectural choices can induce the learning of the entire manifold or a projection of it, where the torus of MLP-Concat is the entire manifold and the vector projection disc-like representation of MLP-Add, Attention 1.0 and 0.0 is a projection.

While our analysis is restricted to modular addition, it illustrates how architectural bias shapes representational geometry, with broader implications for understanding representations and interpreting them. Future work should try to understand why these representations are learned, what aspects of architecture induce representational changes vs. those that don't, as well as whether there's a guiding universal principal that unifies all these representations–because the embeddings and logits are always the same and the vector addition disc is a projection of the torus.

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

# A    Additional details

## A.1    Training hyperparameters.

All models are trained with the Adam optimizer Kingma and Ba [2014]. Number of neurons per layer in all models is 1024. Batch size is 59. Train/test split: $90\%/10\%$.

**Attention 1.0**

- Learning rate: 0.00075
- L2 weight decay penalty: 0.000025

**Attention 0.0**

- Learning rate: 0.00025
- L2 weight decay penalty: 0.000001

**MLP-Add and MLP-Concat**

- Learning rate: 0.0005
- L2 weight decay penalty: 0.0001

## A.2    Constructing representations

In all networks, we cluster neurons together and study the entire cluster at once McCracken et al. [2025a]. This is done by constructing an $n \times n$ matrix, with the value in entry $(a, b)$ corresponding to the preactivation value on datum $(a, b)$. A 2D Discrete Fourier Transform (DFT) of the matrix gives the key frequency $f$ for the neuron. The cluster of preactivations of all neurons with key frequency $f$ is the $n^2 \times |\text{cluster } f|$ matrix, made by flattening each neurons preactivation matrix and stacking the resulting vector for every neuron with the same key frequency.

## A.3    Persistent homology

We compute these using **persistent homology**, applied to point clouds constructed from intermediate representations at different stages of the circuit, as well as the final logits. This yields a compact topological signature that captures how the geometry of these representations evolves across layers, helping us identify when the underlying structure resembles a disc, torus, or circle. We use the Ripser library for these computations Bauer [2021], de Silva et al. [2011], Tralie et al. [2018].

For our persistent homology computations, we set the $k$-nearest neighbour hyperparameter to 250. Our point cloud consists of $59^2 = 3481$ points.

## A.4    Remapping procedure

**Neuron remapping [McCracken et al., 2025a].** For a simple neuron of frequency $f$, we define a canonical coordinate system via the mapping:

$$(a, b) \mapsto (a \cdot d, b \cdot d), \qquad \text{where } d := \left( \frac{f}{\gcd(f, n)} \right)^{-1} \mod \frac{n}{\gcd(f, n)}. \tag{5}$$

This inverse is the modular multiplicative inverse, i.e. for any $\mathbb{Z}_k$ let $x \in \mathbb{Z}_k$. Its inverse $x^{-1}$ exists if $\gcd(x, k) = 1$ and gives $x \cdot x^{-1} \equiv 1 \mod k$. This normalizes inputs relative to the neuron's periodicity and allows for qualitative and quantitative comparisons.

# B    Statistical significance of our results

## B.0.1    Figure 2

We trained 703 models of each architecture, being MLP vec add, Attention 0.0 and 1.0, and MLP concat, and recorded the locations of the max activations of all neurons across all $(a, b)$ inputs to the network. We also computed the center of mass of each neuron as this doesn't always align with the max preactivation (though it tends to be close).

| Description | MMD | p-value | Interpretation |
|---|---|---|---|
| MLP vec add vs Attention 0.0 | 0.0968 | 0.0000 | Moderate difference; highly significant |
| MLP vec add vs Attention 1.0 | 0.1239 | 0.0000 | Clear difference; highly significant |
| MLP vec add vs MLP concat | 0.2889 | 0.0000 | Very strong difference; highly significant |
| Attention 0.0 vs Attention 1.0 | 0.0338 | 0.0000 | Subtle difference; highly significant |
| Attention 0.0 vs MLP concat | 0.1987 | 0.0000 | Strong difference; highly significant |
| Attention 1.0 vs MLP concat | 0.1723 | 0.0000 | Strong difference; highly significant |

(a) Row 1: Max activation

| Description | MMD | p-value | Interpretation |
|---|---|---|---|
| MLP vec add vs Attention 0.0 | 0.0583 | 0.0000 | Small difference; highly significant |
| MLP vec add vs Attention 1.0 | 0.0689 | 0.0000 | Moderate difference; highly significant |
| MLP vec add vs MLP concat | 0.2614 | 0.0000 | Very strong difference; highly significant |
| Attention 0.0 vs Attention 1.0 | 0.0210 | 0.0084 | Subtle difference; highly significant |
| Attention 0.0 vs MLP concat | 0.2126 | 0.0000 | Strong difference; highly significant |
| Attention 1.0 vs MLP concat | 0.1947 | 0.0000 | Strong difference; highly significant |

(b) Row 2: Center of mass

Table 1: Gaussian-kernel Maximum Mean Discrepancies (MMD) Gretton et al. [2012] and permutation p-values between the empirical distributions shown in Figure 2. For each architecture comparison, we sampled 20,000 points from each empirical distribution (derived from histogram-based neuron statistics), then computed the unbiased Gaussian-kernel MMD with a bandwidth chosen via the pooled median heuristic. Significance was assessed using 50,000 permutation tests per comparison.

**B.0.2   Figure 5: Torus distance from the max activation and center of mass to the line $a = b$**

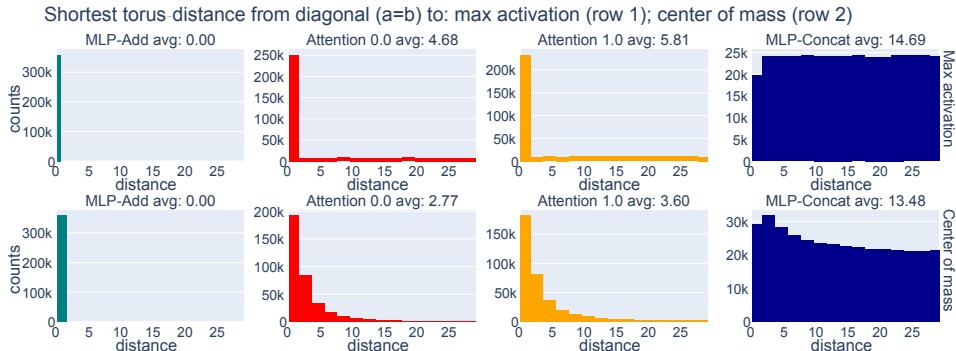

Figure 5: Histograms of torus-distance from each neuron's phase to the diagonal $a = b$, across 703 trained models. MLP-Add neurons align perfectly with the diagonal, Attention 0.0 and 1.0 show increasing off-diagonal spread, and MLP-Concat exhibits broadly distributed activations on the torus.

We trained 703 models of each architecture with 512 neurons in its hidden layer (MLP vec add, Attention 0.0 and 1.0, and MLP concat), and recorded the $a, b$ value of where the max activation of a neuron takes place across all $(a, b)$ inputs to the network and all neurons. We also computed the $(a, b)$ values for the location of the center of mass of each neuron as this doesn't always align with the max preactivation (though it tends to be close). Then we compute the shortest torus distance from the point of the max activation or the center of mass, to the line $a = b$.

Table 2: Gaussian-kernel Maximum Mean Discrepancies (MMD) Gretton et al. [2012] and permutation p-values between the empirical distributions shown in Figure 5. For each architecture comparison, we sampled 2000 points from each empirical distribution (derived from histogram-based neuron statistics), then computed the unbiased Gaussian-kernel MMD with a bandwidth chosen via the pooled median heuristic. Significance was assessed using 5000 permutation tests per comparison.

(a) Row 1: Max activation

| Description | MMD | p-value | Interpretation |
|---|---|---|---|
| MLP vec add vs Attention 0.0 | 0.3032 | 0.0000 | Strong difference; highly significant |
| MLP vec add vs Attention 1.0 | 0.3888 | 0.0000 | Very strong difference; highly significant |
| MLP vec add vs MLP concat | 0.9508 | 0.0000 | Extremely strong difference; highly significant |
| Attention 0.0 vs Attention 1.0 | 0.0705 | 0.0000 | Moderate difference; highly significant |
| Attention 0.0 vs MLP concat | 0.6323 | 0.0000 | Very strong difference; highly significant |
| Attention 1.0 vs MLP concat | 0.5695 | 0.0000 | Very strong difference; highly significant |

(b) Row 2: Center of mass

| Description | MMD | p-value | Interpretation |
|---|---|---|---|
| MLP vec add vs Attention 0.0 | 0.7727 | 0.0000 | Extremely strong difference; highly significant |
| MLP vec add vs Attention 1.0 | 0.7517 | 0.0000 | Extremely strong difference; highly significant |
| MLP vec add vs MLP concat | 0.9148 | 0.0000 | Extremely strong difference; highly significant |
| Attention 0.0 vs Attention 1.0 | 0.0520 | 0.0006 | Moderate difference; highly significant |
| Attention 0.0 vs MLP concat | 0.7022 | 0.0000 | Very strong difference; highly significant |
| Attention 1.0 vs MLP concat | 0.6391 | 0.0000 | Very strong difference; highly significant |

## C  Previous interpretability metrics [Zhong et al., 2023]

### C.1  Definitions

**Gradient symmetricity** measures, over some subset of input-output triples $(a, b, c)$, the average cosine similarity between the gradient of the output logit $Q_{(a,b,c)}$ with respect to the input embeddings of $a$ and $b$. For a network with embedding layer $\mathbf{E}$ and a set $S \subseteq \mathbb{Z}_p^3$ of input-output triples:

$$s_g = \frac{1}{|S|} \sum_{(a,b,c) \in S} \text{sim}\left(\frac{\partial Q_{abc}}{\partial \mathbf{E}_a}, \frac{\partial Q_{abc}}{\partial \mathbf{E}_b}\right)$$

where $\text{sim}(u, v) = \frac{u \cdot v}{\|u\| \|v\|}$ is the cosine similarity. It is evident that $s_g \in [-1, 1]$.

**Distance irrelevance** quantifies how much the model's outputs depend on the distance between $a$ and $b$. For each distance $d$, we compute the standard deviation of correct logits over all $(a, b)$ pairs where $a - b = d$ and average over all distances. It's normalized by the standard deviation over all data.

Formally, let $L_{i,j} = Q_{ij,i+j}$ be the correct logit matrix. The distance irrelevance $q$ is defined as:

$$q = \frac{\frac{1}{p} \sum_{d \in \mathbb{Z}_p} \text{std}(\{L_{i,i+d} | i \in \mathbb{Z}_p\})}{\text{std}(\{L_{i,j} | i, j \in \mathbb{Z}_p\})}$$

where $q \in [0, 1]$, with higher values indicating greater irrelevance to input distance.

### C.2  Results of evaluation

Figure 6 shows the mean and standard deviation of the gradient symmetricity and distance irrelevance metrics from Zhong et al. [2023]. Unlike Zhong et al. [2023], who report gradient symmetricity results over a randomly selected subset of 100 input-output triples $(a, b, c) \in \mathbb{Z}_p^3$, we compute the metric exhaustively across **all** $59^3 = 205,370$ triples to add accuracy.

MLP-Add and MLP-Concat cluster on opposite extremes, implying the metrics just identify whether neurons have phases $\phi_a \neq \phi_b$. MLP-Add models have high gradient symmetricity and low distance

irrelevance and MLP-Concat models have low gradient symmetricity and high distance irrelevance. Attention 1.0 models span a wide range between these extremes depending on two factors: 1) how well the frequencies they learned intersect and 2) how well neurons are able to get their activation center of mass away from the $\phi_a = \phi_b$ line. Attention 0.0 is closer to MLP-Add than Attention 1.0 because it's harder for this architecture to learn $\phi_a \neq \phi_b$. Notably, failure cases exist using both: **neither metric distinguishes between Attention 1.0 and 0.0 models.** While their metrics were intended to distinguish between Clock and Pizza, we show that they are not really able to and this makes sense because Clock and Pizza are not actually different.

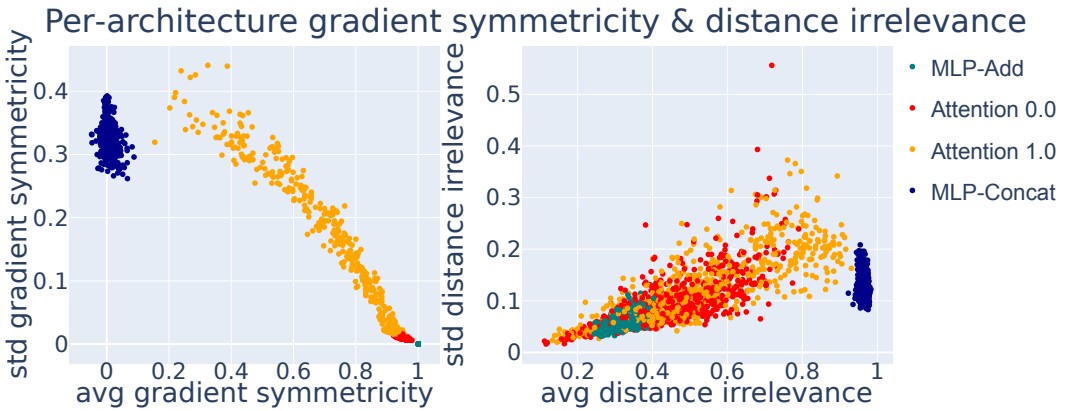

Figure 6: Evaluation of gradient symmetricity (left) and distance irrelevance (right). Each point shows the average (avg) and standard deviation (std) of one trained network. MLP-Add and MLP-Concat lie at nearly opposite extremes, while attention 0.0 and 1.0 overlap substantially. Gradient symmetricity separates Attention 1.0 better, but neither metric **always** distinguishes between Attention 1.0 and 0.0.

### C.2.1 MMD analysis

MMD results for these two metrics are reported below, again showing that the distance between attention 0.0 and attention 1.0 models is small. This is the case even those these metrics were chosen to differentiate between the two architectures.

Using just the x-axis (since the y-axis on those plots is the std dev) MMD results are presented next.

We can conclude that the attention transformers are far from vector addition, and very close to each other under all metrics.

Table 3: Permutation–test MMDs on the empirical gradient symmetricity and distance irrelevance distributions across all architectures. All p-values are $\leq 10^{-6}$ (reported as 0.0000).

(a) Gradient symmetricity (2-D: *avg* and *std*)

| Description | MMD | p-value | Interpretation |
|---|---|---|---|
| MLP vec add vs Attention 0.0 | 1.2725 | 0.0000 | Extremely strong difference; highly significant |
| MLP vec add vs Attention 1.0 | 0.9688 | 0.0000 | Extremely strong difference; highly significant |
| MLP vec add vs MLP concat | 1.3471 | 0.0000 | Extremely strong difference; highly significant |
| Attention 0.0 vs Attention 1.0 | 0.7750 | 0.0000 | Very strong difference; highly significant |
| Attention 0.0 vs MLP concat | 1.3503 | 0.0000 | Extremely strong difference; highly significant |
| Attention 1.0 vs MLP concat | 1.2360 | 0.0000 | Extremely strong difference; highly significant |

(b) Distance irrelevance (2-D: *avg* and *std*)

| Description | MMD | p-value | Interpretation |
|---|---|---|---|
| MLP vec add vs Attention 0.0 | 0.7534 | 0.0000 | Very strong difference; highly significant |
| MLP vec add vs Attention 1.0 | 0.7079 | 0.0000 | Very strong difference; highly significant |
| MLP vec add vs MLP concat | 1.2488 | 0.0000 | Extremely strong difference; highly significant |
| Attention 0.0 vs Attention 1.0 | 0.2078 | 0.0000 | Moderate difference; highly significant |
| Attention 0.0 vs MLP concat | 1.2255 | 0.0000 | Extremely strong difference; highly significant |
| Attention 1.0 vs MLP concat | 1.0990 | 0.0000 | Extremely strong difference; highly significant |

Table 4: Permutation-test MMDs on scatter-plot averages only (1-D). All $p$–values are $\leq 10^{-6}$, so every difference is "highly significant." Note that the distance between attention 0.0, attention 1.0, and MLP vec add is large, implying they are not performing vector addition.

(a) Row 3: Gradient symmetricity (avg only)

| Description | MMD | $p$-value | Interpretation |
|---|---|---|---|
| MLP vec add vs Attention 0.0 | 1.2755 | 0.0000 | Extremely strong difference; highly significant |
| MLP vec add vs Attention 1.0 | 0.9842 | 0.0000 | Extremely strong difference; highly significant |
| MLP vec add vs MLP concat | 1.3833 | 0.0000 | Extremely strong difference; highly significant |
| Attention 0.0 vs Attention 1.0 | 0.7726 | 0.0000 | Extremely strong difference; highly significant |
| Attention 0.0 vs MLP concat | 1.3802 | 0.0000 | Extremely strong difference; highly significant |
| Attention 1.0 vs MLP concat | 1.2559 | 0.0000 | Extremely strong difference; highly significant |

(b) Row 4: Distance irrelevance (avg only)

| Description | MMD | $p$-value | Interpretation |
|---|---|---|---|
| MLP vec add vs Attention 0.0 | 0.7739 | 0.0000 | Extremely strong difference; highly significant |
| MLP vec add vs Attention 1.0 | 0.7268 | 0.0000 | Very strong difference; highly significant |
| MLP vec add vs MLP concat | 1.2501 | 0.0000 | Extremely strong difference; highly significant |
| Attention 0.0 vs Attention 1.0 | 0.2109 | 0.0000 | Strong difference; highly significant |
| Attention 0.0 vs MLP concat | 1.2443 | 0.0000 | Extremely strong difference; highly significant |
| Attention 1.0 vs MLP concat | 1.1093 | 0.0000 | Extremely strong difference; highly significant |

