# OpenReview forum: "The Geometry and Topology of Modular Addition Representations"
_NeurIPS.cc/2025/Workshop/UniReps — UniReps2025_

### Official Review · Reviewer_MSeC · 2025-09-15
**Paper review**

**Confidence:** 2

**Review:**

In this paper, the authors analyze the representations learned on a toy task (modular arithmetic) using either uniform or learnable attention (Pizza and Clock, respectively).
Geometric and topological properties are evaluated using two methods: PAD (Phase Alignment distribution) measuring how often the phases of the learned representations align, while Betti numbers characterize the topological structure of the learned spaces of different layers.
The authors find that the internal representations of these two Pizza and Clock closely align in terms of phase distribution, and their topological trajectories across layers converge in the same way (following different paths).

**Strengths**

- The paper is generally well-written and self-contained, despite the brevity.
- The experimental setup and result seem to be solid.

**Weaknesses**

- The analysis on the topological and geometric properties of the learned representations is interesting, but it is not clear to me how much the results could be extrapolated to other settings.
- Some references in Appendix C are buggy (e.g., Lines 187 and footnote of page 8).

**Score:**

3

**Topic Fit:**

2

---

### Official Review · Reviewer_j5vr · 2025-09-16
**Interesting and Creative Work, but Incomplete in Rationale and Substance**

**Confidence:** 4

**Review:**

## Overall Evaluation

This paper revisits the long-standing “Clock vs. Pizza” debate in modular addition circuits, proposing **Phase Alignment Distributions (PADs)** and **Betti number analyses** as new geometric and topological tools for comparing representations. The main claim is that Pizza and Clock are essentially the same representation under these views, while **MLP-Add** and especially **MLP-Concat** represent genuinely different circuits.

The problem is interesting, and the angle is novel. However, the rationale for the chosen methodologies is underdeveloped, and the connections back to attention mechanisms remain unclear. The amount of work is modest, and while the results are thought-provoking, they do not feel substantial enough even for a short version. I lean toward **weak reject**, though it could be included if space permits.

## Quality and Clarity

**Clarity**: The paper is written clearly, with strong visual support. The exposition makes technical concepts like PAD and Betti numbers accessible to the interpretability audience.

**Quality of Analysis**: The authors provide extensive experiments results. However, the analyses remain descriptive rather than explanatory, and the paper does not justify why PADs and Betti numbers are the right tools to adjudicate Pizza vs. Clock equivalence.

## Originality

*   The use of **PADs** and **Betti numbers** is novel in this context. Applying topological data analysis to neural representations is an original contribution.
*   The idea of showing Pizza and Clock collapse to the same representation is also refreshing, even if somewhat expected given earlier observations of their similarity.
*   Proposing **MLP-Add** and **MLP-Concat** as contrasting baselines adds to the originality, especially since MLP-Concat shows a qualitatively different torus structure.

## Significance

*   **Positive significance**: The paper challenges an influential dichotomy in mechanistic interpretability (“Clock” vs. “Pizza”) and reframes the discussion using geometry and topology. This helps move beyond anecdotal circuit sketches toward more formal metrics.
*   **Limitations**:
    *   The claim that Pizza and Clock are “the same” rests entirely on PADs and Betti numbers. But equivalence under these views does not imply true mechanistic equivalence. Without a stronger theoretical justification, the conclusion feels overstated.
    *   The proposed architectures (MLP-Add, MLP-Concat) are interesting, but the connection back to **attention 0.0 and attention 1.0** is tenuous. The paper never convincingly shows how these new baselines illuminate the role of attention.
    *    The overall scope is modest.

Thus, while the work makes an interesting point, its significance is limited by methodological choices and lack of depth.

## Pros and Cons
1.   (+) **Interesting problem and novel angle**: PADs and Betti numbers provide creative extensions of prior interpretability analyses.
1.   (+) **Insightful result**: The finding that Pizza and Clock collapse to the same representation is provocative and worth documenting.
1.   (+) **New proposal**: MLP-Add and MLP-Concat are substantially different from Pizza/Clock, adding useful contrast.
1.   (-) **Weak rationale for methods**: PADs and Betti numbers are introduced without strong justification. Their equivalence results may reflect limitations of the metrics, not of the models themselves.
1.   (-) **Unclear connection to attention**: The link between MLP-Add/Concat and the original attention-based architectures is underexplored, leaving the framing incomplete.
1.   (-) **Limited contribution**: The paper is primarily empirical, covers only a narrow niche. Even for a short paper, the contribution feels insubstantial.

## Recommendation

I recommend **weak reject**. The paper poses an interesting problem and contributes creative tools, but the methodological rationale is thin, the conclusions are overstated, and the overall contribution is modest. That said, if space allows, it could be included as a stimulating but exploratory piece.

**Score:**

2

**Topic Fit:**

2

---

### Official Review · Reviewer_1KVG · 2025-09-18
**This paper uses geometric and topological analysis to argue that Clock and Pizza circuits are actually learning the same underlying solution.**

**Confidence:** 3

**Review:**

The framework of applying PADs and Bettis numbers is innovative and robust for comparing learned representations. The paper also makes a strong, falsifiable claim that directly challenges and refines influential work in a well-established interpretability testbed.


Questions:
1. One of the most interesting findings is that a model with learnable attention ("Clock") and one with fixed attention ("Pizza") converge to the same representational geometry. Does this suggest that for simple, highly structured tasks like this one, the optimization process effectively "prunes" the flexibility of learnable attention to a simpler, more rigid operation?
2. It would be fascinating to see if other architectural tweaks could produce a genuinely novel third type of representation. For example, have you considered using different activation functions (like GeLU instead of ReLU in the MLP) or various forms of attention (like linear attention)? This could help map out the complete "solution space" for this problem.

**Score:**

4

**Topic Fit:**

2

---

### Official Review · Reviewer_7oVC · 2025-09-19
**Review of modular attention representations**

**Confidence:** 1

**Review:**

I’m going to start by saying that if one were to pick a paper maximally outside of my area of expertise, it would be this paper. Thus, my review will be brief and it’s likely that I missed a few points.

1. One thing I missed in the abstract is why it is relevant to understand these representations.
2. Ln. 15 states ‘our new architecture, MLP-Concat…’, but at the same time McCracken et al. have already analyzed it (Ln 33). Could you clarify the novelty?

**Score:**

3

**Topic Fit:**

3